# Application of Compound-Specific Isotope Analysis in Environmental Forensic and Strategic Management Avenue for Pesticide Residues

**DOI:** 10.3390/molecules26154412

**Published:** 2021-07-21

**Authors:** Eun-Ji Won, Hee-Young Yun, Dong-Hun Lee, Kyung-Hoon Shin

**Affiliations:** 1Department of Marine Sciences and Convergent Technology, Hanyang University, Ansan 15588, Korea; ejwon@hanyang.ac.kr (E.-J.W.); heyun2@hanyang.ac.kr (H.-Y.Y.); Korea; ldh301@korea.kr (D.-H.L.); 2Institute of Marine and Atmospheric Sciences, Hanyang University, Ansan 15588, Korea; 3Marine Environment Research Division, National Institute of Fisheries Science, Busan 46083, Korea

**Keywords:** pesticide, pesticide residues, unintended pollution, stable isotope, compound-specific isotope analysis (CSIA), forensic

## Abstract

Unintended pesticide pollution in soil, crops, and adjacent environments has caused several issues for both pesticide users and consumers. For users, pesticides utilized should provide higher yield and lower persistence while considering both the environment and agricultural products. Most people are concerned that agricultural products expose humans to pesticides accumulating in vegetation. Thus, many countries have guidelines for assessing and managing pesticide pollution, for farming in diverse environments, as all life forms in soil are untargeted to these pesticides. The stable isotope approach has been a useful technique to find the source of organic matter in studies relating to aquatic ecology and environmental sciences since the 1980s. In this study, we discuss commonly used analytical methods using liquid and gas chromatography coupled with isotopic ratio mass spectrometry, as well as the advanced compound-specific isotope analysis (CSIA). CSIA applications are discussed for tracing organic pollutants and understanding chemical reactions (mechanisms) in natural environments. It shows great applicability for the issues on unintended pesticide pollution in several environments with the progress history of isotope application in agricultural and environmental studies. We also suggest future study directions based on the forensic applications of stable isotope analysis to trace pesticides in the environment and crops.

## 1. Introduction

In recent years, concerns about pollution have intensified with high human activity and the increasing use of various chemical products. For example, the disposal of chemicals, including manufacturing chemicals (e.g., plastic additives), personal care products, pharmaceuticals, and pesticides, is a serious threat to various environments [1,2]. Most of these chemicals are toxic and have the potential to accumulate in organisms. Therefore, the monitoring and management of these chemicals is indispensable along with the use of chemicals. In particular, the substances transferred from the used places require monitoring to evaluate their distribution and behavior in the environment and even focus on tracing the origins for prevention or restoration [3]. Among chemical pollution substances, the use of pesticides is widespread and indispensable for protecting crops and other products from harmful insects, weeds, or fungi in agricultural plantations, and forestry [4,5]. The use of pesticides has contributed to a substantial increase in crop yields since the start of the green revolution in the 1960s [6,7]. At the same time, agricultural pollution (e.g., organic pollutants or heavy metals) has long been a concern for its direct linkage to human health. In particular, pesticides have been considered as principal contributors to agricultural production but have accounted for several toxic and metabolite side effects resulting from overuse [4]. Since the 1970s, studies on the toxicity of pesticides and their effects on soil and aquatic ecosystems and human health have greatly expanded, and some persistent organic chemicals, including chlorinated pesticides, dichlorodiphenyltrichloroethane (DDT), and endosulfan have been phased out in the agricultural industry [8]. Globally, these have been replaced by more biodegradable and less toxic chemicals, but contamination by legacy and recent residues still pose environmental and biological problems. Furthermore, it is well-known that even though many alternatives of less toxic substances to highly hazardous pesticides (HPP) are used, they also pose risks [4,9]. For example, chlorpyrifos, an organophosphate insecticide invented as an alternative to the pesticide DDT, is also toxic, and was banned for indoor use after the passage of the Food Quality Protection Act of 1996. A variety of pesticides are still being developed and produced, and these are continuously managed by characteristics such as persistence (e.g., half-life) and toxicity. It is only to be expected that these are manufactured for their toxic effectiveness to specific organisms, including weeds, insects, and diseases. Many studies have demonstrated that the effects of pesticides on biological molecules, tissues, and organs have resulted in acute or chronic disorders in all organisms, including small life forms in soil and water bodies, as well as to humans [4,10,11]. Furthermore, worldwide surveys have documented the toxic effects of these pesticides in humans as they remain in the soil and accumulate in crops, and these pesticides sometimes exceed stipulated health guidelines [12]. However, although pesticides that are relatively ‘safer’ in terms of persistence or toxicity as substitutes are being used, consumers have concerns about the dose and risks because the most current pesticides also affect non-target organisms in a variety of environments, including soils and aquatic systems [4]. These problems not only cause toxicity, but also impact the quality of water, soil, and food for humans, which might lead to health problems in ecosystems [10,13].

In recent years, as the social demand for organic farming and organic crops has increased, issues pertaining to unexpected pollution caused by pesticides contaminating the soil or inflow from outside have also increased [13]. This issue has become a serious problem not only for consumers, but also for users and exporters with strong guidelines and growing interest in organic farming, as well as various pesticide control regulations [14,15]. Consequently, these issues have spurred several agencies globally to develop guidelines and technologies for the safer use of pesticides [16]. In particular, the problem of unintended pollution in diverse systems and products by farmers and other end users remain. Furthermore, notwithstanding the creation of a list of pesticides that can be used depending on the crop type and stipulated concentration, unintended pollution by pesticide residues in soil and/or ground water may still affect crops and must meet stringent requirements for sale. However, despite the extensive use and unexpected contamination in several entities, including water, soil, and even off-season crops, the technology for tracing the source of pesticides associated with prolonged use, degradation, and distribution is rather poorly developed, and uncertainties still exist.

Stable isotope analysis has been introduced as a very useful and novel approach for tracing the source and fate of these chemicals, as the values help to understand the kinetic fractionation of compounds during specific processes, including production (sources), degradation, and other processes occurring in environments with reference to standard values [17,18]. Isotope fractionations, as a result of an enrichment of a lighter isotope relative to another relatively heavier isotope in a chemical or physical process occurring due to the difference of bond energy of each isotope, can provide the information on reactions or its environmental status [17,18]. Stable isotope analysis is a well-established method used in ecological studies and has recently been applied in diverse research fields. Regarding the isotopic signatures of pollutants, several studies have already demonstrated the potential of the isotope approach in environmental forensic sciences that trace the origin and fate of polycyclic aromatic hydrocarbons (PAHs), polychlorinated biphenyls (PCBs), and even metals (Cd, Pb, Zn, and Hg) using stable isotope ratios [19,20,21]. Many scientists have also been interested in exploring stable isotope variability of pesticides as they contain elements (C, N, H, Cl) that can be targeted for measuring during kinetic processes such as degradation, absorption, and desorption mechanisms in soil and water [22,23,24]. Some studies have also reported the mobility and bioavailability of pesticides in water bodies using stable isotope values [25]. However, there has been no attempt to apply stable isotopes to the emerging problems associated with remaining pesticides in soil and tracing the source of pesticide pollution. Furthermore, the topics of previous reviews in the field are still focused on the effects on crops and adjacent environments and monitoring that could be considered in the assessment of its source or impacts of remains in diverse environments (Appendix A).

This paper collates information of stable isotope analyses used under various study fields as an approach for monitoring source tracking. In addition, by understanding the behavior of materials using fractionation of isotopes in compounds, we present a research direction relating to unintended pesticide pollution, which can be adopted and applied in the agricultural field. This study discusses an overview of novel techniques using stable isotope analysis on pesticides and demonstrates the potential of the technology to trace the source and monitor the fate of pesticides in various stages, whilst providing a direction for future research.

## 2. Pesticides and Current Affairs

### 2.1. Several Problems of Pesticides in History and Chemical Control

For several decades, pesticides have become an integral component of agriculture worldwide, which has created a significant increase in the production of crops and food resources [4]. However, it is well-known that pesticides are persistent and have crucial effects such as carcinogenicity, developmental (including teratogenicity in offspring) and reproductive toxicity, mutagenicity, and neurotoxicity effects on various organisms, including plants, aquatic organisms, livestock, and humans [4,13]. This is predictable as pesticides are primarily responsible for controlling fungi and insects. In the Stockholm Convention, twelve persistent organic pollutants (POPs), including nine HPPs (aldrin, chlordane, DDT, dieldrin, endrin, heptachlor, hexachlorobenzene, mirex, and toxaphene) causing severe repercussions on ecosystems and organisms, were banned from being produced and used worldwide [8]. Despite the prohibition of the use of some pesticides in many countries, alternatives are also toxic because they are developed for the purpose of suppressing pests and weeds based on modes of action, such as alteration of enzyme systems in metabolism, neurotransmission dysfunction, oxidative phosphorylation, and electron transfer [26,27]. In particular, studies for establishing standards and monitoring guidelines in environments and evaluating biological impacts are required.

Comprehensive exposure assessments and research evaluating toxicity, degradability, and transformation of products have also been conducted to determine and predict the repercussions of pesticides in use and their degradation processes in soil and water [28,29]. Many studies have reported how these chemicals directly and indirectly affect organisms from molecular to individual levels, as well as in soil and water environments [27,30,31]. Ultimately, these studies are focused on toxicity, fate in diverse ecosystems (e.g., movement and distribution), and accumulation in organisms. These results have been well-organized and published in several reviewed papers (Appendix A). Recently, due to the increase in the quality of food, including crops and livestock products, heightened concerns regarding health and pesticide residues are increasing [16]. Pesticide residues in soil, for example, can be overestimated in terms of use and cause of health and environmental problems.

In fact, the concentrations measured in crops or livestock (e.g., pesticide residues) may not be enough to cause an actual health risk. However, the synergistic effect on health is likely in the case of simultaneous and long-term exposure to two or more pesticides in real-life conditions [11]. In order to reduce these problems, several aspects of pesticide use have to be monitored, such as the process and duration of the pesticides, the composition of pesticides, and the management and sale of products. Several efforts have been made in many countries for the use and management of pesticides. For example, in the U.S., Canada, and Australia, a ‘Zero (or No)-Tolerance Policy’ aimed at managing pesticides at a non-detection level is being implemented [32]. The EU prohibits the sale of foods containing greater than a specified concentration of pesticides for which residual standards have not been established. To establish the target pesticides, Korea and Japan’s Ministry introduced the positive list system (PLS) on all agricultural products in a bill revising part of the Law [33,34]. The PLS permits only certain pesticides and residuals of these pesticides that follow the standard in which the concentration is less than 0.01 ppm in crops [34]. This system was adopted to improve the safety and protection of domestic agricultural products. However, there is still the possibility that high concentrations of pesticides may inflow from outside to cause unintentional pollution.

### 2.2. Emerging Problems of Pesticide: Unintended Pollution to Environments, Crops, and Humans

The use of pesticides is limited to agricultural areas but serves as a non-point source of pollution in a variety of environments [35], since most pollution by pesticides occur through groundwater and spread beyond the points of application. Consequently, several environmental and ecological problems have occurred from residuals, rendering soil and groundwater systems more susceptible to the use of these pesticides [28]. Since the EPA conducted research on pesticide contamination in groundwater in 1979, studies have been intensively conducted and monitoring is being performed by state agencies, pesticide registrants, and universities. As indispensable chemicals in the agricultural sector, discussions on the standards for public health with regard to the use and monitoring of pesticides are ongoing, and new issues are emerging regarding the unintended effects of pesticides [36]. Fipronil is one example, where eggs with significant levels of these herbicides were blocked for sale in the Netherlands, Belgium, Germany, and France after being measured by the Dutch Food and Product Safety Commission. However, several pesticides were found to have accumulated in eggs distributed on the market (pyridaben, fipronil, and even DDT) in Korea. This resulted from contamination of the feed and land where healthy poultry were reared [37]. Following the incident, it was highlighted that the management of pesticide residues was not properly conducted. For agricultural products, it also raised issues such as unintentional contamination due to pesticides remaining in the soil after a long period of time, and when to apply long-term agricultural products such as ginseng, the root of plants in the genus *Panax* [38]. In fact, the pollution of crops or livestock products by pesticides causes environmental and health problems as well as economic losses to the affected people.

The persistence of pesticides has been studied for a long time [4,39,40], however, the accumulation rate of most pesticides has not been found to exceed the range that affects human health until recently, where HPP, such as DDT, has recently been measured in sediments and bivalves of coastal areas [41]. However, the spread or migration of pesticides after use also causes environmental problems, such as discovering and measuring pesticides in areas different from their origin.

Mitra and Raghu [42] confirmed that only a small amount of pesticide affects the target organisms, and most of them come to the plant or soil directly and indirectly. This causes unexpected pesticide contamination in adjacent areas. In fact, in one study, an unexpected high concentration of the organophosphate pesticide diazinon was analyzed even though the use of pesticides was managed and determined in a controlled study site [43]. Spraying is often considered a major factor in adjacent areas where crops with differing residue limits are cultivated in close proximity [43]. Thus, despite efforts to cope with the risks posed by pesticide residues such as PLS, other unintended contaminations are faced, such as residual pesticides detected through air control carried out for forests, contaminations remaining in the soil due to crop rotation, and conversion of other crops.

Many opinions have been continuously raised about the necessity to develop a technique to respond to unintentional pesticide contamination of soil and crops in relation to the use of pesticides [44,45,46]. Soils containing pesticide residues are also a concern with respect to the possible uptake of residues by the following crops. This causes economic losses, as products will be banned from sale. This problem requires the government and farmers to take precautions that differ from those taken in the past for monitoring and management. To manage the unexpected contamination of crops from pesticides, the plant-back interval (PBI) system has been adopted in many countries [47,48], based on the characteristics of pesticides that have half-lives through photochemical and biodegradation processes. The government also presents “guidelines” by building an organization for dispute mediation. In Korea, the management system of unregulated agricultural pesticides is focused on the list of pesticides for sale and use [49].

Despite these efforts, however, unintended pesticide pollution is still considered a sensitive and serious issue in agricultural products. The aquatic environment near agricultural systems is also a target for this serious unintentional pollution [2]. Carvalho et al. [50] showed that several pesticides, including chlorpyrifos, DDT, and parathion, were predominantly found in aquatic systems near agricultural regions (e.g., coffee and leguminous plantation) in a ^14^C-labeled compound experiment. Furthermore, consistent results are also found in central and North American counties these days [51,52]. In addition, many studies have reported that significant levels of pesticides exist in creeks, rivers, and even coastal areas [53,54], indicating predictability for its inflow route, although it is generally considered a non-point source. Furthermore, other studies have been conducted to evaluate pesticide pollution, and the measurement of the natural abundance of stable isotopic ratios of pesticides, and a newly rising technique, is considered a standout option to trace the origins or fate of pesticides to monitor pollution.

## 3. Highlighted Approach: Stable Isotope Analysis

### 3.1. Practical Use of Stable Isotope for Tracing Source

For decades, isotope tracking has been used in a wide variety of fields, from medical to the environment, to understand the processes and pathways of molecules by using the fact that it can be a probe by artificial addition [55,56]. In studies on pesticides, experiments usually involve the administration of isotope-labeled compounds (^14^C) to organisms that are then held in metabolism cages that allow separate collection of expired gases, urine, and feces [57]. It has also been adopted for follow-up studies using ^14^C-labeled compounds (e.g., chlorpyrifos, DDT, and parathion) as markers for the behavior of substances in aquatic environments [50]. Cupples and Sims [58] also showed non-target analysis of degraders by stable isotope probing (SIP) using ^13^C-labeled parent pesticides (herbicide, 2,4-dichlorophenoxyacetic acid (^13^C-labeled 2,4-D)), allowing labeling, isolation, and subsequent amplification of degrader DNA, to demonstrate the biotransformation potential in soil.

The insights gained during these biochemical studies were extended to ecology and its underpinning environmental study because natural isotope values also contain information about the formation and reaction processes of compounds by kinetic reaction between light and heavy isotopes [59,60]. It is easily explained that the elements that constitute a compound have their pristine values and can be altered by preferential reactions in the chemical transformation of lighter isotopes with lower activation energies. For example, C3 and C4 plants, CAM plants, and marine phytoplankton can be distinguished by different stable carbon isotope values of organic matter by different metabolic pathways involved in photosynthesis [61]. This is a simple example showing that the isotope values of elements can be reflected in organisms through specific processes in organisms or in environmental media. The results of ecological studies have shown that the fractionation of carbon-stable isotopes in the diet is not large, though the food web has been applied to find diet sources in the ecosystem [17,59]. Many trials have been conducted using natural stable isotopes to interpret many aspects using specific mechanisms involved in chemical and biochemical reactions in diverse fields, including entomology, fishery science, and ecology [62,63,64]. The stable isotope value is expressed as a delta value (δ), which represents the relative difference of a ratio of the number (or amount) of two isotopes in a sample compared to that of a reference, which is usually an international standard [59]. Therefore, tracing the migration of fish such as the Antarctic salmon *Salmo salar* was possible because the specific elements that compose organic compounds can be affected by independent variables and figures representing their abundance in habitat environments [65]. In a study that targeted particulate organic matter as a source for fisheries, seasonal and individual size-dependent values of stable isotopes of carbon and nitrogen in Manila clam *Ruditapes philippinarum* allowed the interpretation of differences in organic sources that may affect shellfish growth [66].

Recently, this approach, using the intrinsic values of isotopes or patterns of reactions, has been widely used in the establishment of forensic technology (Table 1). It can be concluded that forensic science is an approach to applying scientific knowledge and procedures in criminal investigations to obtain answers to who, what, when, and how to support criminal evidence [67]. In the environment, environmental forensics refers to the application of scientific techniques for identifying the source of compounds or tracking their fate in ecosystems and environments, and has been expanding in academia, government, and commercial markets (commercial products such as wine, sesame oil, and honey) over the past few years [68,69,70,71]. For environmental studies, for example, we can use these isotopic values to trace the anthropogenic source derived from adjacent industrial areas with light values of nitrogen (δ^15^N) by comparing with the concentrations of persistent toxic substances according to the distance of sources in our recent study [72]. This is an easy and effective approach for discriminating various factors with different inflow distributions in time and confirming the contribution rate by designating potential sources, such as rural, urban, and industrial. Interestingly, this approach is being used not only for alive ‘modern’ samples, but also applied for ‘ancient’ samples to understand socioeconomic status and life history in archeology [73,74,75]. Archeology has also been applied to ancient samples, helping to interpret the past. Cooper et al. [75] distinguished the ratios of the stable isotopes of C, N, and S in modern human hair with differing economic practices (farmers, pastoralists, fishers) based on dietary differences. Subsequently, the stable isotopes of organisms and compounds are a ‘Logbook’ of the target objects. In the field of ecology and fisheries, stable isotope ratios of multiple elements also provide information on the geographical origins and help to understand the habitat. It is directly related to improving or monitoring the environment for the purpose of economic gain and is also useful in solving social problems such as fraudulent fishery products of geological origin.

### 3.2. Applications of Compound-Specific Isotope Analysis (CSIA) to Pollutants in Environmental Sciences

In general, a variety of compounds are routinely used to analyze their isotopic compositions (C, N, H, etc.) at natural abundance. The isotope analysis of these compounds is conducted via offline and online combustion systems combined with IRMS (Figure 1). Novel analytical techniques with gas chromatography-isotope ratio mass spectrometry (GC-IRMS) provide stable isotope ratios of several elements (C, N, H) of organic compounds, including contaminants in environmental samples. The CSIA approach has the unique potential to elucidate the transformation mechanisms of organic pollutants directly in the environment with the manner and order of chemical bond cleavage generated in chemicals [19,20,60,76]. In the late 20th century, there was an attempt to analyze compound isotopes for substances used as various geological and biogeochemical indicators, including natural (e.g., fatty acids and alkanes) and anthropogenic (e.g., PAH and volatile organic compounds such as benzene, toluene, ethylbenzene, and xylene (BTEX)) compounds [19,87,88]. Compared with the conventional methods of analyzing isotope composition of carbon, nitrogen, etc., in total organic matter (e.g., bulk isotope analysis), CSIA can be used to trace the source of specific substances using the stable isotope values of the components that are involved in chemical reactions, including the generation or breakage of chemical bonds of organic contaminants [89]. These initiatives aim to find the source of pollutants and interpret the distributed biological movements in relation to their behavior in the environment, which continues to this day. Therefore, several analytical challenges associated with the low concentrations and high polarity of micropollutants have been attempted by combining this method and have shown great potential for understanding mechanisms in environments and filling in the knowledge on their origins [19,20,60,76]. This is based on the different isotope values of different geographic sources [19].

For relatively low molecular weight volatile compounds such as MTBE, BTEX, or chlorinated solvents, the isotopes, primarily carbon and hydrogen, have been used extensively to evaluate the onset of natural attenuation, while for larger molecules such as PCBs or PAHs, in which the effects of biodegradation on the isotope composition of these molecules are minimal, the isotopic fingerprints of the individual compounds can be used for correlation purposes [90]. Furthermore, in the first review of the CSIA, Schmidt et al. [91] provided examples of successful CSIA applications in environmental studies, including (i) the allocation of contaminant sources on a local, regional, and global scale, (ii) the identification and quantification of (bio)transformation reactions on scales ranging from batch experiments to contaminated field sites, and (iii) the characterization of elementary reaction mechanisms that govern product formation, and even mentioned that carbon analysis has already reached the ‘maturity stage’.

**Figure 1 molecules-26-04412-f001:**
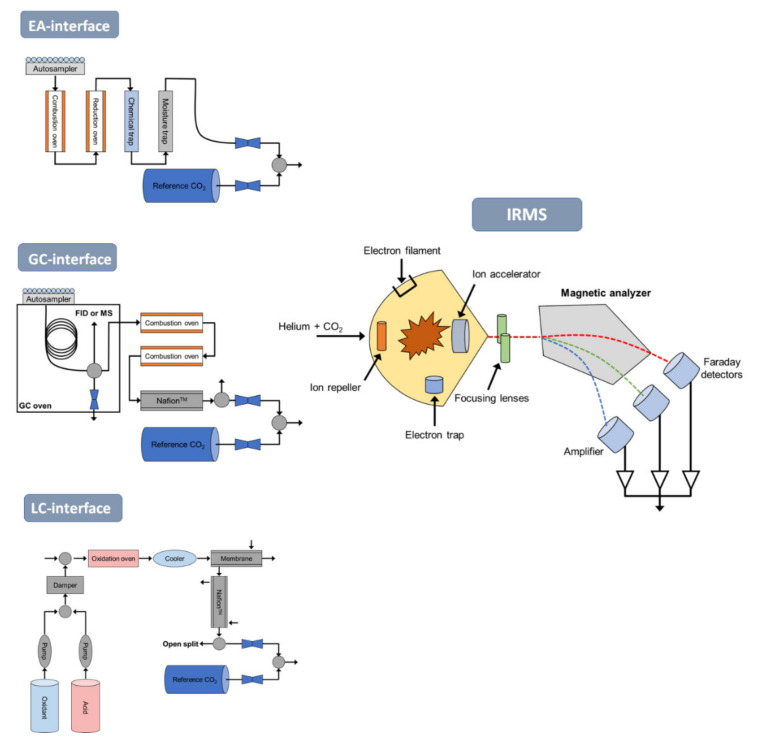
Measurement of isotope ratios by isotope ratio mass spectrometry (IRMS) combined with: (A) Elemental analysis-isotope ratio mass spectrometry (EA-IRMS), in which solid samples are combusted and separated into gases by an elemental analyzer before being introduced into the IRMS. (B) Gas chromatography-combustion-isotope ratio mass spectrometry (GC-C-IRMS), in which samples are injected into a gas chromatograph for separation prior to combustion and introduction into the isotope ratio mass spectrometer, and (C) liquid chromatography-isotope ratio mass spectrometry (LC-IRMS), in which samples are injected into a liquid chromatograph for separation and then flow into an isotope ratio mass spectrometer. Figure was slightly modified from Muccio and Jackson [92] (adapted with a permission from ref. [92]. Copyright 2009, The Royal Society of Chemistry).

PAHs are one of the compounds that scientists are interested in determining source apportionment and finding the fate, with a well-discussed endmember of carbon isotope ratio (δ^13^C) [93,94]. Each compound from low (e.g., naphthene, acenaphthylene, acenaphthene, etc.) to high molecular weight (e.g., pyrene) has its own δ^13^C values according to its generation, such as C3 plant combustion, wood combustion, diesel exhaust, domestic coal combustion, and petrogenic sources, etc. [94]. This information can be incorporated into the mathematical isotopic model for calculating the contribution of mixed sources. In a recent study, Yang et al. [94] reported that 43.9% of Shanghai’s roadside agricultural soil was contaminated with a major contribution from traffic sources, which was identified as the δ^13^C value of PAH from fossil fuels. This approach can be directly used for biological samples. In a study on the transport of pollutants in a food chain and its biological metabolism, analysis of ^13^C isotope values of PCB congeners and PBDE confirmed the significant correlations between the sediment and the fish species and between the fish, indicating the same origins of these pollutants [80]. The advantage of measuring the isotopes of contaminants has allowed various instruments to be combined and technical tests to be attempted. For aqueous and volatile organic contaminants, optimal results were obtained with a purge-and-trap interface to the GC-IRMS instrument [95]. Recently, several developments, such as combination with the bacterial denitrification method with gas bench and the trace gas concentration system in IRMS with GC, have allowed nitrogen isotope composition of nitrate [78,96].

The application of CSIA to pesticides is akin to the measurement of the isotopes of PAHs or other organic pollutants in environments, such as source tracing, understanding the fate, and providing information within ecosystems (bioaccumulation). When it comes to pesticides, the primary sources of pesticides in the environment are agriculture and forestry, but the occurrence of pesticide residues, called unintended pollution by diffusion and bioaccumulation, is measured in small amounts and in degraded forms. This is why what is measured in the environment is often perceived as a non-point source, making it difficult to hold users accountable. It also explains the need for research on tracing the source of pesticides and several attempts to date using CSIA.

## 4. Applications of CSIA for Monitor Pesticide Pollution in Environments and Agricultural Products

### 4.1. Pesticide Tracing Using Isotope of Parental Compounds

Isotope analyses have been considered an important technique in almost all areas of pesticide research for selective toxicity, persistence, and delivery using traceability (e.g., probing) [97]. Once again, this technique can contribute not by tracing, but a behavior study using natural values. With respect to forensic sciences, the application of CSIA has been adopted to determine the source of materials or pollutants and provide novel concepts for future directions of pesticide monitoring in soil, groundwater, watersheds, and crops. This is based on the principle that raw materials have different isotope values even if the major ingredient is the same, and it is also possible that the isotope values of these constituents may differ due to the conditions of the manufacturers’ processes [25]. For example, methomyl is synthesized by reacting *S*-methyl *N*-hydroxythio-acetimidate (MHTA) in methylene chloride with gaseous methyl isocyanate over a wide range of temperatures (−30 to 50 °C), and each manufacturer has inherent raw materials and optimum processes. In the study by Dreznek et al. [22], the stable isotope ratios of Cl and C of twelve Aroclors from different suppliers showed differences, and researchers demonstrated that the source materials or the process of synthesis, purification, or post-production storage and handling of the products may be responsible for their different values, although a correlation between the synthetic pathway and the δ^37^Cl was observed.

Generally, gas chromatography-combustion-isotope ratio mass spectrometry (GC-C-IRMS) has been used to measure the isotope values of the main ingredient in pesticides. This system is suitable for separating pesticides from pesticide metabolites. It improves the detection of the isotope values of specificity in the compound of interest. A recent study by Song et al. [84] also showed the great potential of using stable isotopes to trace the source of methomyl in murder cases caused by methomyl as poison, which is unaffected by methomyl metabolite (i.e., methomyl oxime) using GC-IRMS. A methomyl, a prohibited insecticide, is highly toxic and is often used for self-poisoning and sometimes for poisoning of murder convicts [98,99]. In 2016, in the process of investigating deaths from drinking alcohol with added methomyl, a forensic research team was able to find a suspect with a pesticide with the same carbon-stable isotope value of methomyl in the alcohols. The stable value of the methomyl carbon isotope served as a ‘smoking gun’ to find suspects, and this study also presented results on fractionation in the process of obtaining evidence, such as no changes in isotope ratio on the influence of intestinal pH [84]. These interesting examples support the possibility of the differentiation of pesticides through differences in isotopes in raw materials and composition.

In another study of CSIA of pesticides, the findings also suggested that using a combination of several isotopes (e.g., C, N, and Cl) could provide greater certainty in determining the origin of a given herbicide [35]. Subsequently, several cases have been reported that successfully locate or trace the source of pesticides using isotopic ratios not based only on pesticide [43,84]. This allows stable isotopes to play a very important role in determining the origins of the raw materials; therefore, a study that analyzed eight different diazinon manufacturers, and extensively used organophosphate insecticide, showed that values of δ^13^C were significantly different in each product, ranging from −30.7‰ to −21.7‰. This study showed great possibilities for tracing the source of diazinon by showing the comparative results between soil from the study area surrounded by paddy fields of different farmers and eight different products. Using a similar range of isotopes, the results suggested that the two pesticides were most likely responsible for the contamination [43]. These studies are based on a non-degraded fraction of pesticides and assume that there would be no isotope changes.

### 4.2. Isotopic Fractionations in Degradations of Pesticides

Until recently, attempts of the CSIA approach to pesticides had a consistent purpose, which was to investigate pesticides in the environment and trace their sources and fate. However, unlike previous studies that focused on the stability of pesticides, except in the process and storage (Section 4.1), the pesticide compounds used in crops undergo dynamic physical, chemical, and biological processes, including volatilization and chemical and biological degradation. Degradation, a substantial process, is a property that decreases after acting on the target object and is also the key feature that has allowed the use of pesticides safely. It is also the fundamental process for attenuating pesticide residue levels in the environment [100,101]. However, the degradation changes isotopic values of pesticides and influences the potential of the isotopes as tracers. These isotopic fractionations are the result of the processes in which lighter isotopes can react faster or be preferentially incorporated in one step, indicating that isotopic studies can be used to understand the fate of chemicals. As a result, the product becomes isotopically lighter, and the source reactant is heavier. With regard to pesticides, the favored transformation of lighter isotopes (e.g., ^12^C) during a degradation process results in an enrichment of heavier isotopes (e.g., ^13^C) in the residual substrate fraction (Figure 2). Isotope fractionation in pesticides occurs via the cleavage rate of the molecule’s bonds through various reactions [24,45].

The natural degradation of pesticides to various reactions in the environment and the resulting change in the isotope ratio vary significantly. However, this happens in a predictable manner, so it is believed that the stable isotope analysis is still applicable for the characterization of degradation pathways and reaction mechanisms of pesticides, although compounds are degradable, and their isotope values are flexible [60]. Many recent studies have suggested that the CSIA, which focuses on one compound, may provide a complementary line of evidence to tackle pollutant degradation in environmental systems at different temporal and spatial scales by quantifying or specializing [25,89,102,103,104]. Briefly, insights into the impact of the substituent position on the apparent ^13^C, ^15^N, and ^37^Cl (apparent kinetic isotope effects (AKIEs) of a specific isotope) are considered as a key for environment monitoring as the information on mechanisms of bond cleavage could be obtained [24]. We collated the studies that focused on finding isotopic fractionation under several experimental conditions, such as analytical procedures and decomposition under various conditions (Table 2). The studies presented in Table 2, together with isotopic fractionation values of products (metabolites) or residues from various degradation processes, represent the potential of the CSIA approach for future studies. Furthermore, these studies have highlighted consistency in findings, but understanding the mechanisms of reactions is another challenge for their application to the environment, and their importance and helpfulness in interpreting environmental samples.

For example, photolysis is one of the processes many studies have been interested in [105,106,107]. The reaction rate altered by environmental factors can also be studied using isotopic analyses. In a study by Wu et al. [107], the differential isotopic fractionations of parathion (an organophosphate insecticide) during hydrolysis at different pH values (C-O cleavage causes εC −6.9~−6.0‰ at pH 2–7, while no fractionation of δ^13^C at pH > 12 (e.g., P-O cleavage)) suggested that the AKIEs allow a deep interpretation of hydrolysis (e.g., photolysis) processes, which is needed for future environmental monitoring.

Biodegradation (e.g., biotransformation, bacterial hydroxylation) occurring in the environment has also been studied in pure culture experiments or lab-scale catchment studies [24,108]. In particular, the mechanisms of biological degradation are well-discussed in previous studies. In a study by Meyer et al. [109], for example, different fractionations of δ^13^C and δ^15^N in the reaction of atrazine to hydroxyatrazine were observed in two different species and discussed as connected to enzymes involved in the transformation pathway (AtzA vs. TrZN) (dual isotope slopes Δ (=δ^15^N/δ^13^C) for *Chelatobacter heintzii* (−0.65 (±0.08)) and *Arthrobacter aurescens* TC1 (−0.61 (±0.02)). Kuntze et al. [93] also asserted that anaerobic biodegradation is distinguished from aerobic biodegradation and abiotic transformation in most organic pollutants from dozens of results using ME (multi elements)-CSIA. This is because the cleavage site that affects fractionation differs depending on environmental conditions. Thus, studies showing these comparative results suggest that the initial step of biodegradation pathways in environments can be addressed based on the difference in fractionation.

The relationship between these AKIE concepts and chemical reactions, such as decomposition, can be directly applied to plants that can be used as prime targets for pesticides. In a recent study on hexachlorocyclohexane (HCH, lindane), the increasing δ^13^C signature of HCH combined with the enantiomeric fraction (EF) value of α-HCH (an index of preferential degradation of α-HCH) could explain the degradation process of these compounds in the rhizosphere of plants, although the results still have limitations in determining the contribution of biodegradation in the rhizosphere and within plants (soil–plant system) [110]. Concepts related to the quantification of the decomposition of pollutants have been discussed using the ‘Rayleigh equation’ approach, that informs the change in the isotopic enrichment of the degraded substrate and allows the calculation of the isotopic composition of the product, since the 2000s [104].

Owing to findings and the potential of residues as the only informative evidence in environmental applications to trace residue origins, many studies have investigated how these data can be applied to environmental monitoring in waters and sediments [45,111]. Research on this topic has shown some examples that were measured in water samples from cultured systems as alternative evidence of contaminant degradation extent in the environment or some environmental samples [29,44]. Alvarez-Saldivar et al. [29] applied mass balance to evaluate the natural attenuation of pesticides at the catchment scale. The stable isotope sources and sinks model is one of the examples developed by Lutz and Breukelen [112] using a mathematical model for two mixing sources and degradation via one reaction pathway. This model is based on a linear stable isotope mixing model and the Rayleigh equation and can cover the simultaneous occurrence of degradation processes and mixing of emission sources of organic pollutants and even inorganic compounds such as nitrate. All of these studies demonstrated that the use of compound stable isotope ratios opens the possibility of tracing different origins for regulating environmental pollution by interpreting the time of use and the process of decomposition in the environment, as it is generally independent of non-destructive dissipation (e.g., dilution, sorption, and volatilization). However, although the calculations for source apportionment and the quantification of the extent of degradation were in good agreement with the simulation results, the large uncertainties in the CSIA data remain, and the unknown interplay between competing reaction pathways that occur in environments presents difficulties in interpretation and can result in uncertainties.

**Table 2 molecules-26-04412-t002:** Overview on CSIA studies conducted on pesticides (herbicide, fungicide, and insecticide). Almost all studies are from the last 20 years or so (since 2002). These are focused on natural abundances of isotopic compositions, except SIP. OC and OP indicated in the target pesticide list indicate organochlorine and organophosphate compounds, respectively.

Investigated Pesticide	Elements (CSIA)	Study Objectives or Target Mechanisms (for Isotopic Fractionation)	Sample Types (or Field Site)	Special Features	References
Hexachlorocyclohexane (HCH, Lindane) (OC, insecticide)	C (α-, γ-HCH)	- Reductive dechlorination (Sulfate reducing bacteria)	- Lab experiment (pesticide chemicals itself)	- HCH dechlorination can be monitored in anoxic environments	[113,114]
	C (α-HCH)	- Biotransformation (Dehalococcoides species)	- Enrichment culture	- The first ESIA study for understanding the biodegradation process of α-HCH	[115]
	C (α-HCH)	- Direct and indirect photolysis, alkaline hydrolysis, electro-reduction and reduction by Fe (0)	- Lab experiment (pesticide chemicals itself)	- The study showed discriminate results in chemical and biological transformations.	[106]
	C (α-, β- HCH)	- Fate of HCH in environments	- Field sample (sediment, waters, cow milk, plants, etc.)	- The first study using the combination of isotope fractionation and EF to trace the reactive transport processes in the environment, including food web	[110]
	Cl, C	- Isotopic values of each compound	- Pesticide product	- The first study applying isotopes, and has strength to provide a baseline for future work employing isotope ratios to study the environmental fate of SVOCs (semi-volatile organochlorine compounds)	[22]
Chlordene (OC, insecticide)	Cl, C	- Isotopic values of each compound	- Pesticide product		
Heptachlor (OC, insecticide)					
Dieldrin (OC, insecticide)					
Aldrin (OC, insecticide)					
Mirex (OC, insecticide)					
Hexachlorocyclopentadiene					
Cypermethrin (class II pyrethroid, insecticide)	C	- Isotopic values of different production	- 16 products	- The study shows the possibility as a forensic tool in criminology	[116]
Isoproturon (Phenylurea, herbicide)	C, N, H	- Hydrolysis - Biodegradation - Fungal/bacterial—hydroxylation	- Lab experiment (pesticide chemicals itself) - Pure culture experiments	- Studies showed quantifying contaminant degradation using isotope and also showed biotransformation and photolytic reactions of the phenylurea herbicide.	[117,118]
Atrazine (triazine, herbicide)	C, N, H	- Photolysis (254 nm)	- Lab model system (Surface waters)	- The study provided isotope fractionations that can be used in environments for determining phototransformation pathways.	[105]
	C, N, H	- Experimental condition - Alkaline hydrolysis	- Lab experiment (pesticide chemicals itself)	- The study found conditions for accurate and precise isotope values in its procedure and provided fractionation of C and N at alkaline hydrolysis.	[23]
	C, N, H	- Biodegradation	- Lab experiment (pesticide chemicals itself)	- Different transformation occurred in biotic and abiotic experiments. The mechanistic insights on enzymatic reactions were discussed.	[109]
	Cl	- Biodegradation	- Pure culture experiments	- The first study on the fractionation in Cl	[119]
	C, N	- Abiotic hydrolysis	- Pesticide standards	- Significant isotopic fraction was measured at alkalic condition (pH 9).	[45]
	C, N	- Biodegradation	- Bacteria with atrazine	- Found fractionation for bioavailability	[120]
	C, N	- Extraction method	- Drainage waters	- Study suggests solid-phase extraction with consecutive clean-up by HPLC	[46]
	C	- Extraction method	- Ground waters	- This study showed the fractionation in an extraction procedure and mentioned the careful method evaluation of sample preparation and sample pretreatment prior to reliable CSIA.	[121]
2,6-dichlorobenzamide (benzamides, herbicide)	C	- Extraction method	- Ground waters		[121]
	C, N	- Fractionations in mesoscale aquifer	- Aqueous samples	- Application study to monitor the reactivity of micropollutants in aquifers and guide future efforts to accomplish CSIA at even lower concentrations	[122]
Diazinon (OP, Insecticide)	C	- Source tracking	- Eight diazinon products and paddy soils	- Carbon isotopes values of diazinon narrow down the candidate products	[43]
Alachlor, Acetochlor and Metolachlor (Chloroacetanilide, herbicide)	C, N	- Biodegradation	- Pure culture experiments	- Provide assessing method on the fate of the metabolite (BAM) of pesticide in environments	[108]
	C, N	- Degradation	- Lab-scale wet land samples	- The first study on isotopic enrichment revealed by biodegradation of chloroacetanilides	[24]
	C, N	- Abiotic hydrolysis	- Drainage water	- The study showed different fractionations of pesticides in pH conditions.	[45]
	C, N	- Extraction procedure in method	- Tap water/drainage water	- This study validated the SPE approach for large samples to analyze low concentrations of aquatic pesticides.	[46]
S-metolachlor/Acetochlor (Chloroacetanilide, herbicide)	C	- Degradation (field)	- Catchment	- The first study applying isotopes to assess pesticide transformation at catchment scale using a conceptual model	[44]
Glyphosate (OP, herbicide)	O (water)	- UV degradation	- Lab experiment (pesticide chemicals itself)	- Confirmation of cleavage through oxygen isotope measurement using EA-gas bench-CF-IRMS	[123]
	C, N	- Analytical procedure (derivatization)	- Lab experiment (pesticide chemicals itself)	- Showed high precisions of isotope values in LC-IRMS approach	[124]
Dimethoate (OP, insecticide)	C, H	- Photolysis/hydrolysis	- Aqueous sample	- The study for developing a method for the analysis of carbon isotope signatures of three OP pesticides	[111], [125]
dichlorobenzene isomers (insecticide)	C	- Anaerobic degradation	- Lab experiment (pesticide chemicals itself)	- Result showed small isotopic fractionation in anaerobic biodegradation and the limitation of CSIA to quantify in situ biodegradation.	[126]
Parathion (OP, insecticide)	C	- Hydrolysis - Photolysis	- Lab experiment (pesticide chemicals itself)	- The study showed different fractionations in hydrolysis of parathion according to pH conditions.	[107]
Metalaxyl (Acylalanine, fungicide)	C, N	- Abiotic hydrolysis	- Pesticide standard	- Significant isotopic fraction was measured at alkalic condition (pH 9).	[45]

### 4.3. Analytical Method and Trials for Residual Pesticides

Table 3 summarizes notable studies where high precision was obtained in isotope analyses for various compound characteristics and analytical techniques. In general, offline systems perform compound isolation and subsequent offline combustions for the analysis of isotopic compositions. On the other hand, online systems achieve continuous flow of GC (or LC)-combustion in IRMS platforms, and allow on-column sample injection and chromatographic separation of target compounds to be converted into gas form (CO_2_, N_2_, and NO_x_) in a combustion reactor. Accordingly, compared to offline systems, online systems require relatively lower sample volumes. It is particularly important to analyze substances of interest at low environmentally relevant concentrations of compounds.

GC/IRMS is useful for analyzing the isotope composition of non-polar organic compounds. Alternatively, a proper derivatization procedure can be introduced to change the polarity of the target compounds into less reactive forms [127,128]. LC/IRMS is also suitable for analyzing samples that contain polar and/or nonvolatile compounds [129]. This method helps to obtain high precision and accuracy, as it does not require a derivatization step that is prone to errors caused by the addition of carbon, which is required in conventional GC/IRMS [127]. For instance, carbon isotope analysis of glucose and ethanol was assessed by LC/IRMS to develop criteria for identifying brewers’ sugars and alcohols by applying the binding method to separation, purification, and pretreatment [129]. This method was suggested as it required only a short time to measure and a small amount of sample. However, in recent years, with improvements in commercial high-performance liquid chromatography, sufficient quantities of target compounds and their metabolites have been isolated for isotope analyses. The potential of conventional LC isolation of polar compounds and subsequent offline isotope analysis has been re-evaluated (e.g., amino acids, in Sun et al. [127]).

The CSIA is also applicable to pesticides (e.g., herbicides, fungicides, and insecticides), and its applications are still emerging, which is due to the fact that the sample amount required is much higher to obtain accurate isotope values in specific compounds than in conventional quantitative analysis. In fact, a large volume of sample or more than 100,000-fold pre-concentration is required [46]. Thus, the pre-concentration step of pesticides in environmental samples and agricultural products should be performed with a heavy workload. For the study of pesticide residues and metabolites in trace concentrations (ng/L to µg/L range), recent studies have focused on pre-concentration procedures to obtain a sufficient volume of highly clean extraction for isotope analysis [46,128,130]. Thus, there are concerns that pretreatment methods, such as solid-phase extraction (SPE), might cause isotope fractionation during isotope measurement [43,128,130], although several studies have shown the stability of SPE as a pretreatment process, with negligible changes in stable isotope ratios in some pesticides [24,46]. Melsbach et al. [128] tested SPE sorbents, such as graphitized carbon-based to polystyrene-divinylbenzene, to validate multiclass herbicides for routine isotope analysis (Table 3). However, despite these diverse trials from environmental water samples, little information is available on complex biological matrices [131]. This might be due to the matrix complexity in samples where pesticide residues are extracted with other compounds at the same time [121]. A previous study suggested that most classes of pesticides show a wider scope and better sensitivity if detection is based on an LC-based platform for their quantification [132]. However, there are only few analyses of pesticides in the analytical platform using LC/IRMS for pesticides when compared to analyses using GC-IRMS (Table 4). LC/IRMS was used to measure the carbon isotope value in the analyte, while GC/IRMS can detect both C and N isotope compositions. The limited capacity of LC/IRMS is caused by analytical limitations to quantitatively differentiate N_2_ produced from N-containing compounds against the high background of N_2_ in air. LC amendable compounds are derivatized by adding C from a derivatizing reagent (e.g., methylation) and allowed for GC analysis. However, carbon isotope analysis using LC/IRMS can provide more accurate values of pesticides and their metabolites because it does not require a derivatization step in sample treatment. Inadvertently, this is because derivatizing carbon(s) on the analyte can shift the isotope values of the pesticide as well as its metabolite, and thus isotope correction procedures should be accompanied by handling isotopic data. This indicates that LC/IRMS might be useful for tracking low concentrations of pesticides in the environment in terms of C isotope analysis.

Recently, multi-isotope analysis (C, H, N, and Cl) in pesticides has been developed to competitively investigate the source and transformation routes in the environment [110]. Several studies have suggested that considering more than one (e.g., C, N, Cl) isotope variable clearly improves the discrimination power of a pesticide derived from different suppliers [35,128,133]. In recent studies, δ^13^C and δ^15^N signatures of desphenylchloridazon (metabolites of herbicide chloridazon) in water were obtained by LC/IRMS and by derivatization GC/IRMS to obtain multi-isotope variables [128]. Mogusu et al. [125] also reported isotope fractionation of dual elements (C and N) of glyphosate and its metabolite aminomethylphosphonic acid (AMPA) during abiotic oxidation to obtain information to trace the sources and degradation of glyphosate using GC/IRMS combined with LC/IRMS. Dual column coupling (e.g., GC-GC/IRMS) also represents a more attractive option for obtaining high resolution and separation capacity in the C isotope analysis of analytes in complex matrices [134].

If specific isotope analytical methods of pesticides have adopted qualitative GC and LC systems that quantitatively analyze parent compounds and metabolites, they may not be ideal to produce reliable isotope data [121]. Torrentó et al. [46] mentioned that LC/IRMS methods are more susceptible to interferences that could compromise the accuracy of isotope analyses produced by the concomitant enrichment of organic matrix compared to GC-IRMS methods. Thus, using an LC-based platform across multi-class pesticides can be considered as an area that requires considerable research in the future. Furthermore, it is crucial to develop a pre-concentration, purification, and separation protocol for isotope values in diverse pesticide classes to interpret the process that can occur during sorption and transformation in environments and agricultural products (e.g., Reference [128]). Finally, the fact that most studies have been conducted in water samples, whereas sediment samples and agricultural product samples have not been studied, suggests the need for further studies regarding in situ samples and agricultural products.

**Table 3 molecules-26-04412-t003:** Examples of studies that developed or used a new intermediate course for IRMS analysis for pesticides and other compounds.

Target Compounds	Conventional Platform	Objectives (Purpose of the Method Development)	Samples	Additional Procedures	Platform	Strength Points of New Method	References
Amino acids (Methionine)	GC- IRMS	For using source amino acids combined with other source AA (Phe)	23 terrestrial and coastal organisms	Purification of methionine isolation (HPLC-CAD)	GC-IRMS	Measures low concentrations of Methionine	[134]
Amino acids	GC-IRMS GC-IRMS	For high precision	Standard and squid	Purification (Multidimensional HPLC)	EA-IRMS	No derivatization	[127]
Ethanol and glucose	EA-IRMS (IC/LC extraction and purification)	To develop the criteria to check brewers’ alcohol and sugar	Sake (Alcohol)	–	LC-IRMS	Less sample volume, Fast analysis	[129]
Pesticide—Atrazine	–	To find conditions for accurate and precise isotope values, the types of combustion reactor tube fillings and the effects of Ni/NiO furnace on N isotopes were tested.	Pesticide itself	–	GC-IRMS	Combustion fillings	[23]
Pesticide—Atrazine and desethylatrazine	EA-IRMS	To measure the isotope values from a large volume of samples having low concentrations	Groundwaters	SPE and HPLC cleanup	GC- IRMS	HPLC reduces matrix effects, and increases sensitivities by on-column injection	[130]
Pesticide—Alachlor, Acetochlor	EA-IRMS	To compare the results of each method and additional extraction procedure efficiency	Waters (Lab-scale wetland water)	SPE extraction procedure	GC- IRMS	High accuracy	[24]
Organophosphate Pesticide—Dichlorvos, Omethoate, Dimethoate	EA-IRMS	To develop the method for pesticide from aqueous samples	Aqueous sample	SPE	GC- IRMS	–	[111]
Pesticide metabolites—Desphenylchloridazon (metabolite of herbicide chloridazon)	GC-IRMS	For the development of methods applicable to small amounts of environmental samples	Agricultural drainage water	Solid-phase extraction (SPE) approaches	LC-IRMS	Isotope ratios can be accurately measured at analyte concentrations in the sub-μg L^−1^ range	[46]
Pesticide—Atrazine, Desethylatrazine, 2,6-dichlorobenzamide (ng/L))	GC-IRMS	To get a high precision for low-concentration water samples	Ground waters	SPE with consecutive clean-up by HPLC	GC-IRMS	High accuracy of isotope values from a large volume of water samples by SPE	[121]

**Table 4 molecules-26-04412-t004:** Representative examples of general methods for residues and measurement systems for analyzing isotopes of pesticides in agricultural products and environmental samples.

Pesticide (and Its Metabolite)	Residual Quantification Platform	Platform for CSIA (Target Elements)			Reference
		EA-IRMS	GC-IRMS	LC-IRMS	
Atrazine	LC-MS/MS ^a,b^, LC/MS	O	O	O	[23,35,45,46,105,109,120,121,130]
Desethylatrazine	SPE-LC-MS/MS ^a^		O		[122]
Desphenylchloridazon ^†^	GC-MS/MS ^a^		O	O	[46]
DDT	GC-MS/MS ^b,c^		O		[81]
Heptachlor	GC-MS/MS ^a,b,c^		O		[22]
Aldrin	GC-MS/MS ^a,c^		O		[22]
2,6-dichlorobenzamide	GC-MS/MS ^a^		O		[122]
Diazinon	GC-MS/MS ^a,c^, LC-MS/MS ^b^	O	O	O	[43,45,46]
Dimethoate	LC-MS/MS ^a,c^	O	O		[111]
Parathion	GC-MS/MS ^a,c^, LC-MS/MS ^b^		O		[107]
Chlordene	GC-platform	O			[22]
Dieldrin	GC-MS/MS ^b^	O			[22]
Mirex (Dechlorane)	GC-MS/MS ^b^	O			[22]
Hexachlorocyclopentadiene (HCCPD) ^§^	GC-platform	O			[22]
Cypermethrin	GC-MC/MS ^b,c^		O		[116]
Isoproturon	LC-MS/MS ^c^		O		[117,118]
Alachlor, Acetochlor, Butachlor, and Metolachlor, S-metolachlor	GC-MS/MS ^a,c^, LC-MS/MS ^c^		O		[24,29,44,45,108]
Chloroacetanilide	LC-MS/MS ^d^		O		[24]
Glyphosate	LC- and GC-MS/MS	O	O	O	[123] ^e^ [124] ^f^
Hexachlorocyclohexane (HCH, Lindane)	GC-FID, -MS/MS	O			[22]
			O		[106,110,113,114,115,132]
Dichlobenil (2,6-Dichlorobenzonitrile)	GC-platform		O		[122]
Acylalanine (Metalaxyl)	LC-MS/MS ^c^	O	O		[45]

^a^ The information on the analytical platform for each compound based on the technical note of Agilent [135]. ^b^ The information on the analytical platform for each compound based on Almeida et al. [136]. ^c^ The information on the analytical platform for each compound based on Rathod et al. [137]. ^d^ The information on the analytical platform for each compound based on EPA [138]. ^e^ EA-gas bench-IRMS was used for measuring stable oxygen isotope ratios of glyphosate. ^f^ The stable isotope ratios of nitrogen and carbon of glyphosate were analyzed by GC and LC based IRMS, respectively. ^†^ Desphenylchloridazon is a metabolite of chloridazon. ^§^ Hexachlorocyclopentadiene is the key intermediate in the manufacture of some pesticides, including heptachlor, chlordane, aldrin, dieldrin, and endrin.

## 5. Implications

In the review by Philp [90], environmental forensics have shown possibilities as a research discipline directed toward determining parties liable to cause spills of contaminants into the environment. Recently, the effects of pesticides on crops and organisms have been a cause for concern because of the unintended pollution by chemicals remaining or diffusing in the environment. However, extensive research on biological responses has only focused on reporting the effects of the remaining pesticide chemicals in crops. Numerous studies using novel techniques, such as the CSIA approach, have shown their potential for assessing pesticide inputs and degradations in a variety of systems. We expect that these recent attempts can model the prediction of transformation mechanisms under different environmental conditions and systematically compare possible reaction mechanisms during degradation. This might fill the knowledge gap in the current field for further applications and also give rise to various technological advances in protocols. With respect to the quantitative estimation of pesticides in watersheds, the integrated framework can be supplemented to make precise forecasts for the distribution of pesticides, and this could assist to bring about changes in various government policies and environmental goals for sustainable development.

## Figures and Tables

**Figure 2 molecules-26-04412-f002:**
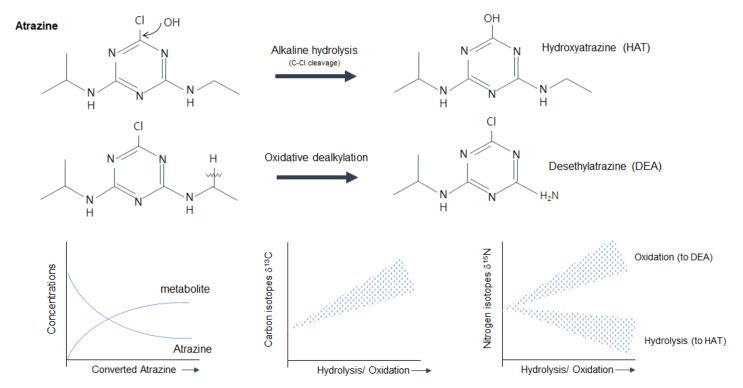
Atrazine concentrations and its degradation products (hydroxtatrazine and desethylatrazine), and corresponding δ^13^C (bottom left) and δ^15^N values (bottom right) during hydrolysis and dealkylation. The figure on the far left shows the concentrations of atrazine and its metabolites, and the *Y*-axis of the two figures on the right shows the carbon and nitrogen stable isotope ratios of atrazine observed differently as atrazine is metabolized (*X*-axis), respectively.

**Table 1 molecules-26-04412-t001:** Representative studies using stable isotope ratios of diverse analytes for forensic sciences. Examples of pesticides only listed studies on the parent compound (pesticide products).

Study Field	Objectives of Analysis	Target Compound	Elements	Highlights	Reference
Environmental	- To define sources of pollutants	- Organic matter	C, N	- Stable isotope values of carbon and nitrogen were clearly distinguished between anthropogenic sources in an artificial lake	[72]
		- PAH	C (PAHs)	- A study found the source of PAH in urban lake sediments	[76]
		- Oil	C	- The results of the aromatic fraction and the isotopic type curves showed the source of oils	[77]
		- PM2.5 (air samples)	C, N (NO_x_), S	- Result showed that coal combustion-related isotopic patterns increased during China’s winter heating period.	[78]
		- Aerosol (Sulfate)	S, O	- The mean δ^34^S value of aerosol sulfate is similar to that of coal from North China, indicating that coal combustion is a significant contributor to atmospheric sulfate.	[79]
		- Pesticide	C	- Carbon isotope values of 8 different products were compared with the ^13^C values of diazinon in paddy soil	[43]
		- PCB	C	- Found correlations between PCB and PBDE in sediment and fish and showed the origins and its transfers in aquatic environments using δ^13^C values of each compound	[80]
		- DDT (in soil)	C (o,p’-DDT)	- Interpret the results of DDT inflow through air transport	[81]
Criminology	- To distinguish diet and habitat characteristics of the corpse	- Tissue/bone/hair	C, H	- Relations between diet/product and geographic location made it possible to identify the murder victim or crime	[82]
	- To find distribution route of marijuana	- Product	C, N	- The source of the sample is classified through the results showing differences in the cultivation environment, such as indoor growth	[83]
	- To get the information of criminal evidence (poison, crime tool, etc.)	- Poisons (methomyl)	C (methomyl)	- Identifying the matching isotope values among the chemicals held by the accused.	[84]
Archeology	- To reconstruct dietary information from paleo-people (bone) and understand social and socioeconomic status and life history	- Bone	N (collagen-amino acid)	- The trials to evaluate freshwater resource consumption using amino acid nitrogen isotope values	[73]
		- Micro-debris in dental calculus	C, N	- The study showing direct evidence of fish and plant consumption in Mesolithic Mediterranean	[74]
		- Hair	C, N, S	- Shows the dietary form and economic situation that is distinct in this region of Ethiopia	[75]
Ecology	- To reconstruct food source, trophic guild, and energy chain in target ecosystem	- Tissue	C, N	- Effect of species invasion on food web structure and trophic level, understood as a stable isotope ratio	[17]
	- To find origin (habitat) and migration of wildlife	- Tissue	C, N, S, H, Sr	- Integrated research on the mobility of wild animals using spatially reflected isotope signatures based on various biogeochemical processes	[62]
Food Science	- To prevent food fraud by distinguishing geographical and source origin (fish/ shellfish/livestock product/honey/tea)	- Tissue/Shell	C, N, H, O, S (Sr, Nd)	- Studies have shown results that differentiate:	
				- between wild and farmed fish	[85]
				- between natural honeys and sugar-fed honeys	[71]
				- values of Clams from three different countries	[86]
Fishery	- To find suitable or economically favorable natural aquaculture conditions	- Tissue	C, N	- Found changes in preferred diet according to the growth of manila clams observed in culture and field samples	[66]

## Data Availability

Not applicable.

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
