# Peer review of "Application of Compound-Specific Isotope Analysis in Environmental Forensic and Strategic Management Avenue for Pesticide Residues"

_molecules, 2021, doi:10.3390/molecules26154412_

Round 1

Reviewer 1 Report

Compound-Specific Isotope Analysis (CSIA) is an analytical method that measures the ratios of stable isotopes and has been widely applied as an environmental forensic tool including, but not limited to, illicit drugs, explosives, drug control, and contaminated groundwater studies etc. In other words, the results generated through the CSIA technique work as an important auxiliary tool in decision-making and in understanding the environmental context. The interpretation of the data generated by the CSIA technique depends, among other factors, on the evaluation of the processes that influence isotopic fractionation. Natural processes of biotic and abiotic degradation can cause changes in isotopic ratios, and those that most influence isotopic fractionation are those of transformation, as biodegradation. The CSIA technique has become an important tool in several fields of science. I conclude that the work does reach the level necessary to be accepted for publication in this prominent journal. My comments are as follow:

The authors need to standardize the information regarding the pesticide endosulfan, which is a persistent compound: p. 2, line 49: “...some persistent organic chemicals, including chlorinated pesticides, dichlorodiphenyltrichloroethane (DDT) and endosulfan...”; p. 5, line 215: “... showed that endosulfan, a non-persistent compound, was...”.

page 6, Table 1: the authors could select another term to designate the third column of Table 1, since the term 'analyte' is not entirely adequate.

Table 2 and Table 4: It is necessary to correct the spelling of the word in the text ‘hexachlorocyclopentadiene’

Topic 4.3, line 526: It is necessary to correct the presentation of the reference. ‘[128, 129 Melsbach et al 2019]’

Author Response

First, thank you for your careful comments.  According to reviewer's comments, we thoroughly revised our manuscript and incorporated changed as suggested by reviewer.
Comments from Reviewer 1 and responses
The authors need to standardize the information regarding the pesticide endosulfan, which is a persistent compound: p. 2, line 49: “...some persistent organic chemicals, including chlorinated pesticides, dichlorodiphenyltrichloroethane (DDT) and endosulfan...”; p. 5, line 215: “... showed that endosulfan, a non-persistent compound, was...”.
Response: Thanks for the thoughtful review. We corrected information on chemicals cited in the manuscript. It was a typo that occurred while editing a sentence before submission.
page 6, Table 1: the authors could select another term to designate the third column of Table 1, since the term 'analyte' is not entirely adequate.
Response: Thank you. We changed the ‘analyte’ to the ‘target compound’.
Table 2 and Table 4: It is necessary to correct the spelling of the word in the text ‘hexachlorocyclopentadiene’
Response: Thank you. We correct the spelling of the chemical.

Topic 4.3, line 526: It is necessary to correct the presentation of the reference. ‘[128, 129 Melsbach et al 2019]’
Response:  The form of the references in the text has been thoroughly revised. Thank you for your helpful comments.

Reviewer 2 Report

Review of the manuscript

In Molecules:

Application of compound-specific isotope analysis in environ-2 mental forensic and strategic management avenue for pesticide residues

By Eun-Ji Won, Hee Young Yun, Dong-Hun Lee and Kyung-Hoon Shin

The paper describes current state of knowledge of the compound specific isotope analyses, with focus on tracing the pesticide degradation molecules.

The paper is well written, and in my opinion it can be published, after some minor changes:

-define delta notations, and use it in the text rather than isotope values of…, or change to ratio of stable isotopes of (an element)

-abbreviated words should have their full name

-define isotope fractionation

I also attach the PDF with concrete suggestions on rephrasing some sentences, and points to some type errors.

I only had comments for the manuscript and not in the supplementary material document.

Author Response

We greatly appreciate critical comments of reviewers on our work and thank them for the valuable suggestions on the manuscript. We answered one by one to the comments below and to the comments you gave in the pdf file.

Reviewer 2.

The paper is well written, and in my opinion it can be published, after some minor changes:

-define delta notations, and use it in the text rather than isotope values of…, or change to ratio of stable isotopes of (an element)

- We added the information on delta notations in the revised manuscript as follows.

“The stable isotope value is expressed as a delta value (δ), which represents the relative difference of a ratio of the number (or amount) of two isotopes in a sample compared to that of a reference, which is usually an international standard [59].”

-abbreviated words should have their full name

Abbreviation and full name of the chemicals were carefully checked and corrected in the text. Thank you.

-define isotope fractionation

We added the sentence for explain isotope fractionation in the revised manuscript.

“Isotope fractionations, a result of an enrichment of lighter isotope relative to another relative heavier isotope in a chemical or physical process occurring due to the difference of bond energy of each isotope can provide the information on reactions or its environmental status [17,18].”

“These isotopic fractionations are the result of the processes in which lighter isotopes can react faster or be preferentially incorporated in one step, indicating that isotopic studies can be used to understand the fate of chemicals.”

I also attach the PDF with concrete suggestions on rephrasing some sentences, and points to some type errors.

Thank you for your comments. We totally revised the manuscript as reviewer suggested. As reviewer suggested we have added information to the text on several definitions (delta, Rayleigh equation, ) used in the field of isotopes.